# Towards Understanding Associative Knowledge in Vision-language Models via Neuron-level Attribution

## Abstract

We investigate how the vision encoder of vision-language models (VLMs), such as CLIP, store associative knowledge about specific entities. We develop attribution methods to identify "knowledge neurons" within CLIP's visual encoder that enable recognition of entities such as celebrities, cartoon characters, and cultural symbols. Our analysis reveals that recognition of specific entities is primarily facilitated by a small subset of neurons in the later feed-forward network (FFN) layers. We then propose techniques to dissect these knowledge neurons from both visual and linguistic perspectives, demonstrating that they are activated exclusively by visual signals of specific entities in complex images and encode semantically relevant concepts. Building on these findings, we propose two practical applications: selectively removing sensitive knowledge and inserting new entity associations without degrading overall model performance. Our work contributes novel methods for neuron-level attribution, interpretable techniques for knowledge understanding, and effective approaches for targeted knowledge editing in VLMs.

## 1 Introduction

Vision-language models (VLMs) (Zhang et al., 2024) are multi-modal representation learners that map inputs from multiple modalities into a shared embedding space. A notable example is CLIP (Radford et al., 2021), which is trained on a large collection of image-text pairs using contrastive learning. The visual representations learned by CLIP demonstrate strong performance on various downstream tasks, such as classification (Saha et al., 2024; Gong et al., 2025a), segmentation (Wang et al., 2022; Yu et al., 2023), and object detection (Lin & Gong, 2023; Vidit et al., 2023). Additionally, several variants of CLIP have been developed, including SigLIP (Zhai et al., 2023), MetaCLIP (Xu et al., 2024), and EVA-CLIP (Sun et al., 2023). They are also widely used as the vision encoders of MLLMs.

Beyond its remarkable performance, the CLIP model demonstrates impressive multi-modal capabilities and zero-shot learning potential. For instance, CLIP assigns high similarity scores between image representations and the text representations of their corresponding labels. Interestingly, previous research has shown that, beyond common objects (e.g., vehicles and furniture), CLIP can recognize specific entities (Goh et al., 2021), such as celebrities, cartoon characters, and cultural symbols. It is also capable of identifying the names of highly specific or non-obvious objects, often outperforming MLLMs (Geigle et al., 2024). Notably, the names of these specific entities are often not directly linked to the objects themselves, requiring domain-specific knowledge to establish the connection. For example, recognizing a photo of Snoopy as "Snoopy" rather than simply "a white dog" demands such knowledge (Fig. 1). While it is evident that CLIP possesses the knowledge to identify specific entities, understanding of how this knowledge is encoded within its visual encoder remains limited.

To gain a deeper understanding of how CLIP stores associative knowledge, we develop methods to analyze its internal structure, with a focus on CLIP-ViT (Dosovitskiy et al., 2021). The CLIP-ViT has several properties. First, it employs residual connections, meaning that the output is a cumulative sum of the outputs from individual layers. Additionally, prior research has shown that the neurons within each transformer layer can be interpreted as key-value memory units (Geva et al., 2021; Nichani et al., 2025; Yu & Ananiadou, 2024a), allowing the layer output to be further broken down into value memories. Lastly, CLIP encoders map images and text into a shared representation space, enabling

Figure 1: **Overview of this study.** We propose a framework to identify knowledge neurons within the visual encoder of VLMs, which exhibit associations with specific entities when analyzed from both visual and linguistic perspectives. These findings allow us to efficiently edit knowledge in VLMs.

attribution of individual neurons to specific text concept alignment. As CLIP-ViT and its variants are widely used as visual encoders in MLLMs, this focus also help reveal the associative ability of larger models. By isolating the visual encoder, we avoid the ambiguity that arises when analyzing the LLM component, where it is often unclear whether recognition of a specific entity stems from the encoder's learned associative knowledge or from the LLM's reasoning mechanisms.

Specifically, given a photo of a specific entity, we use CLIP-ViT to encode it into a visual representation. To analyze this process, we propose a variant of the logit lens (Nostalgebraist, 2020), which unembeds intermediate representations into the output space to trace how the network progressively aligns the visual representation with the text embedding of the entity's name. Unlike the original logit lens, which outputs token distributions, our method uses cosine similarity with the text embedding as the metric. We further use the increase in cosine similarity as an importance score for each neuron, allowing us to identify neurons that significantly contribute to the final visual-text alignment.

We collect a dataset consisting of diverse types of specific entities, including celebrities, cartoon characters, logos, landmarks, and films. Through analyzing these entities, we identify a unique type of neurons, referred to as *knowledge neurons*, which are integral to recognizing these specific entities. Each entity is associated with one or a small number of particular knowledge neurons that play a significant role in aligning images with their corresponding text labels. These neurons are primarily located in the final few layers of the feed-forward network (FFN).

To further explore knowledge neurons, we propose methods to dissect and analyze them. From a visual perspective, we can generate saliency maps that highlight the critical parts of images responsible for activating the knowledge neurons. From a linguistic perspective, we can unproject the neuron's value memory into the text representation space, enabling interpretation in plain text. Our findings reveal that the saliency maps precisely identify the specific entities within images, even in scenarios with multiple similar items (e.g., identifying a celebrity in a group of people). Additionally, the text interpretations correspond to concepts directly related to the specific entities (see Fig. 1).

Based on the observations, we propose two applications of the knowledge neurons regarding knowledge editing. By removing the first-order contributions of the knowledge neurons, we can compel CLIP model to forget certain sensitive information. Through rank-one model editing, we are able to insert new knowledge enabling the model to recognize a new specific entity. Notably, these editing will not degenerate the generalization of CLIP. Our contributions can be summarized as follows:

- We introduce a novel neuron-level attribution and dissection framework to detect and interpret the knowledge neurons, providing insights from both visual and linguistic perspectives.

- Experiments conducted on a dataset containing diverse specific entities confirm the existence of knowledge neurons and shed light on their special behaviors.

- We propose two applications of knowledge neurons: removing sensitive knowledge and inserting new knowledge, and show the efficacy of the methods through numerical studies.

## 2 RELATED WORK

**Mechanistic Interpretability.** Mechanistic interpretability (Lin et al., 2025) seeks to reverse-engineer the computational processes of neural networks. It has been widely applied to interpret large language models (LLMs) (Bricken et al., 2023; Conmy et al., 2023; Huben et al., 2024; Bhaskar et al., 2024; Wu et al., 2025). In the field of computer vision, this exploration can be categorized into three levels: neuron-level, component-level, and architecture-level. At the neuron level, researchers aim to understand the specific concepts associated with individual neurons (Bau et al., 2017; Hernandez et al., 2022; Kalibhat et al., 2023; Oikarinen & Weng, 2023a; Ahn et al., 2024; Bai et al., 2024; Shi et al., 2025). At the component level, prior studies have identified network components that fulfill specific functions (Bau et al., 2020; Schubert et al., 2021). At the architecture level, research has focused on understanding how vision transformers process visual information (Pan et al., 2024; Zeng et al., 2024; Wang et al., 2025b). Additionally, mechanistic interpretability has been used to study how visual information flows within VLMs (Schwettmann et al., 2023; Huo et al., 2024; Neo et al., 2025; Yu & Ananiadou, 2024b; Wang et al., 2025a; Xu et al., 2025). In our work, we use mechanistic interpretability to identify and analyze knowledge neurons in VLMs, focusing on the vision encoder rather than the LLM component emphasized in prior work, as vision encoder is the main component where visual features are mapped into semantic space and associative knowledge forms.

**Interpretations of VLMs.** The interpretations of VLMs can be categorized into concept-based and saliency map-based explanations. Concept-based explanations (Chen et al., 2023; Moayeri et al., 2023; Oikarinen et al., 2023; Yang et al., 2023; Yun et al., 2023; Bhalla et al., 2024; Chattopadhyay et al., 2024) aim to break down visual embeddings into multiple concepts represented by text embeddings. Saliency map-based explanations (Li et al., 2022; 2025; Wang et al., 2023; Zhao et al., 2024; Zhu et al., 2025; Gong et al., 2025b) generate saliency maps on input images conditioned on a given text prompt, highlighting image regions most relevant to the text. While most studies focus on understanding the decision-making process of VLMs, only a few explore their mechanistic interpretability. Gandelsman et al. (2024; 2025) decompose attention heads into text-interpretable directions to analyze their roles, and Goh et al. (2021) investigate the behavior of multi-modal neurons in CNN-based CLIP models. Our work does not aim to develop general-purpose VLMs interpretation methods, but designs exclusive teniques to analyze the associative knowledge of VLMs.

**Knowledge Neurons.** The concept of knowledge neurons was first introduced in the context of LLMs to refer to specific components responsible for storing factual information. This idea was initially proposed by Dai et al. (2022), who demonstrated that factual associations are often localized within specific MLP neurons. Building on this, Meng et al. (2022a) developed a framework to identify and edit factual associations in LLMs. Subsequent research (Geva et al., 2022; Nanda et al., 2023; Shi et al., 2024; Yu & Ananiadou, 2024a; Pan et al., 2025) introduced new methods to locate knowledge neurons across multiple layers and attention heads. Yao et al. (2024) further expanded this concept by proposing the idea of knowledge circuits. While these studies have focused exclusively on knowledge neurons in LLMs, the role of knowledge neurons in ViTs or VLMs remains unexplored. Our work pioneers the investigation of associative knowledge neurons in VLMs.

## 3 LOCATING KNOWLEDGE NEURONS VIA ALIGNMENT IMPROVEMENTS

We begin with a brief overview of ViT (Dosovitskiy et al., 2021) and CLIP (Radford et al., 2021) (Sec. 3.1). Then, we introduce our proposed algorithms for locating knowledge neurons (Sec. 3.2) and present techniques for dissecting and interpreting the identified knowledge neurons (Sec. 3.3).

### 3.1 CLIP-VIT PRELIMINARIES

ViT is a residual network composed of $L$ layers, where each layer consists of a Multi-Head Self-Attention (MSA) mechanism followed by an FFN block (Vaswani et al., 2017). The input image $I$ is first divided into $N$ non-overlapping patches, which are linearly projected into $N$ $d$-dimensional vectors. Positional embeddings are then added to these vectors, forming the image tokens $\{h_i^0\}_{i \in \{1, \cdots, N\}}$. Additionally, a learned token $h_0^0 \in \mathbb{R}^d$, referred to as the class token, is included and later serves as the output token. With the residual connections, the output of layer $l$ can be expressed as:

$$h_i^l = h_i^{l-1} + A_i^l + F_i^l, \tag{1}$$

where $A_i^l = \text{ATTN}_i^l(h_0^{l-1}, \cdots, h_N^{l-1})$ represents the output of the MSA block, and $F_i^l = \text{FFN}^l(A_i^l + h_i^{l-1})$ represents the output of the FFN block[1]. This process is repeated for $L$ layers. Finally, the output token corresponding to the class token, $h_0^L$, is used as the final output of the ViT.

In CLIP, a text encoder is used alongside the vision encoder, with both working together to project images and text into a shared representation space. We use $M_{\text{image}}(I)$ and $M_{\text{text}}(T)$ to denote the visual and text representations. The visual representation is obtained by projecting the class token $h_0^L$ through a projection head $\mathbf{P}$, such that $M_{\text{image}}(I) = \mathbf{P}h_0^L$. During training, CLIP employs a contrastive loss to maximize the similarity between an image representation and its corresponding text representation. This similarity is defined as the cosine similarity between $M_{\text{image}}(I)$ and $M_{\text{text}}(T)$. In the inference phase, CLIP encodes image and text into their respective representations, and calculates their cosine similarity. A high similarity indicates that the text is closely related to the image. This capability is particularly powerful for tasks such as zero-shot classification and image-text retrieval.

## 3.2 NEURON-LEVEL ATTRIBUTION

We further break down the outputs of the MSA block and the FFN block. The FFN processes the input by applying a nonlinear activation function $\sigma$ between two linear transformations.

$$F_i^l = \mathbf{W}_{fc2}^l \sigma(\mathbf{W}_{fc1}^l(h_i^{l-1} + A_i^l)), \tag{2}$$

where $\mathbf{W}_{fc1}^l \in \mathbb{R}^{d_f \times d}$ and $\mathbf{W}_{fc2}^l \in \mathbb{R}^{d \times d_f}$ are two weight matrices. Inspired by prior work (Geva et al., 2021), we notice the output $F_i^l$ is essentially a weighted sum of the columns of $\mathbf{W}_{fc2}^l$. The weight for the $k$-th column is the inner product between the $k$-th row of $\mathbf{W}_{fc1}^l$ and the input $h_i^{l-1} + A_i^l$, passed through the nonlinear function $\sigma$. We refer to the pair $(\mathbf{W}_{fc1,k,:}^l, \mathbf{W}_{fc2,:,k}^l)$ as the $k$-th neuron of the $l$-th FFN layer, denoted as L$l$F$k$. The first element is called query vector and the second is value vector[2]. Similarly, the attention output $A_i^l$ can also be expressed in the following matrix form:

$$A_i^l = \sum_{j=1}^{H} \sum_{p=0}^{N} \alpha_{i,j,p}^l \mathbf{W}_{o,j}^l \mathbf{W}_{v,j}^l h_p^{l-1}, \quad \alpha_{i,j,p}^l = \text{softmax}(\mathbf{W}_{q,j}^l h_i^{l-1} \cdot \mathbf{W}_{k,j}^l h_p^{l-1}), \tag{3}$$

where $H$ is the number of attention heads, $\mathbf{W}_{o,j}^l \in \mathbb{R}^{d \times d/H}$, $\mathbf{W}_{k,j}^l \in \mathbb{R}^{d/H \times d}$, $\mathbf{W}_{q,j}^l \in \mathbb{R}^{d/H \times d}$, $\mathbf{W}_{v,j}^l \in \mathbb{R}^{d/H \times d}$ are the output, key, query, and value matrices. The key and query matrices are used to compute the attention weights $\alpha_{i,j,p}^l$ on the $p$-th position through the softmax function. We notice the attention output can also be represented as the weighted sum of the columns of $\mathbf{W}_{o,j}^l$. Similarly, we refer to the pair $(\mathbf{W}_{v,j,k,:}^l, \mathbf{W}_{o,j,:,k}^l)$ the $k$-th neuron of the $j$-th head of the $l$-th MSA layer, denoted as L$l$A$j$H$k$, with the first and the second element to be the query and value vectors.

After defining the concept of a neuron, we proceed to estimate each neuron's contribution to the recognition of a specific entity. Specifically, we define the contribution to be the improvement in visual-text alignment caused by the neuron. Given an image of a specific entity $I$ with text label $T$ and text representation $M_{\text{text}}(T)$, the improvement caused by an FFN neuron L$l$F$k$ is then defined as:

$$\text{Imp}(\text{L}l\text{F}k) = \cos(\mathbf{P}(h_0^{l-1} + A_0^l + m_F^l(k)\mathbf{W}_{fc2,:,k}^l), M_{\text{text}}(T)) - \cos(\mathbf{P}(h_0^{l-1} + A_0^l), M_{\text{text}}(T)), \tag{4}$$

where $m_F^l(k) = \sigma((\mathbf{W}_{fc1,k,:}^l)^T(h_0^{l-1} + A_0^l))$. Similarly, for an MSA neuron L$l$A$j$H$k$, we have:

$$\text{Imp}(\text{L}l\text{A}j\text{H}k) = \cos(\mathbf{P}(h_0^{l-1} + m_A^l(j,k)\mathbf{W}_{o,j,:,k}^l), M_{\text{text}}(T)) - \cos(\mathbf{P}h_0^{l-1}, M_{\text{text}}(T)), \tag{5}$$

where $m_A^l(j,k) = \sum_{p=0}^{N} \alpha_{0,j,p}^l (\mathbf{W}_{v,j,k,:}^l)^T h_p^{l-1}$. Intuitively, the alignment improvement metric evaluates how much incorporating a neuron's contribution enhances the alignment with the text label during the progressive processing of visual features. We refer to a neuron as a *knowledge neuron* if it plays a significant role in recognizing a specific entity, i.e., after including it, the cosine similarity between the visual features and the text features increases significantly.

---

[1]For simplicity, we ignore the layer normalization layer in the derivation. We will explain it in the Sec. A.1.
[2]Note that these terms are unrelated to the query and value components in attention mechanisms.

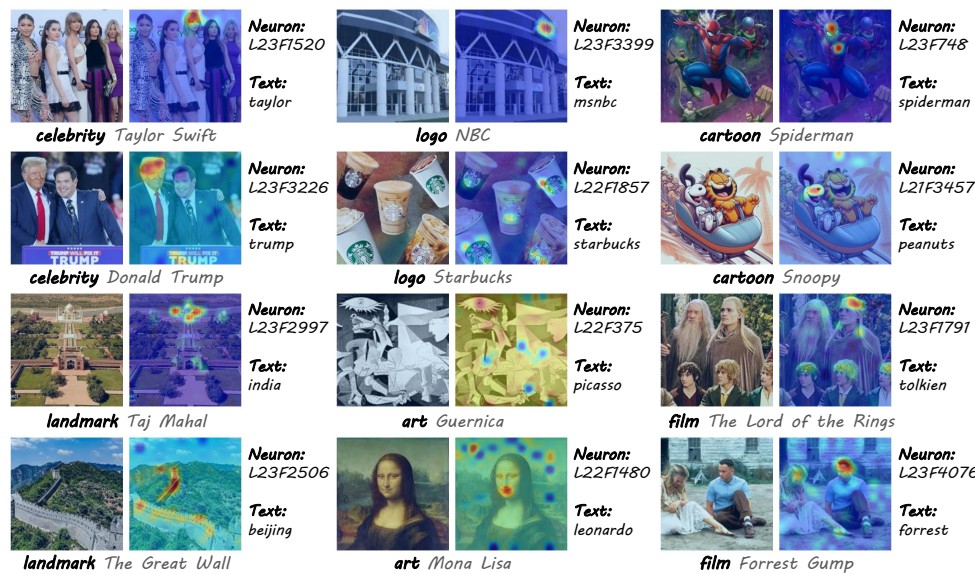

Figure 2: **Illustrations of knowledge neurons and their interpretations.** We present examples of knowledge neurons associated with different types of entities, along with their activation maps on input images and the top textual decomposition results for these neurons.

Note that similar to many existing studies (Elhage et al., 2021; Gandelsman et al., 2024; Yu & Ananiadou, 2024a), we focus only on the first-order effects of each neuron. This means we consider the *direct* flow from the neuron through the residual stream to the output. While this is a simplification of the complex ViT architecture, it has demonstrated strong empirical results in practice.

### 3.3 DISSECTING THE KNOWLEDGE NEURONS

After identifying the knowledge neurons associated with a specific entity, the next step is to interpret the knowledge stored within these neurons. We propose two techniques for interpreting knowledge neurons, approaching the task from both linguistic and visual perspectives.

**Linguistic Perspective.** We observe that the final visual representation is essentially a weighted sum of the value vectors from all neurons. After the projection $\mathbf{P}$, the visual representation is mapped into a shared representation space with the text representation. Consequently, the value vectors of each neuron are also positioned in this same representation space and can be interpreted based on the directions of the text representations. To achieve this, we propose decomposing the value vector into human-understandable concepts. Specifically, for a given value vector of a neuron, $\mathbf{W}^l_{fc2,:,k}$, we first project it into the joint representation space as $\mathbf{P}\mathbf{W}^l_{fc2,:,k}$. Next, we construct a dictionary consisting of common concepts (e.g., the 20k most frequently used English words) and use the text encoder to encode these concepts into text representations, denoted as $\{c_1, \cdots, c_M\} \in \mathbb{R}^d$. We then apply sparse decomposition algorithms, such as orthogonal matching pursuit (Pati et al., 1993), to decompose the value vector into a sparse weighted sum of concept representations: $\mathbf{P}\mathbf{W}^l_{fc2,:,k} = \sum_{m=1}^M \omega_m c_m + \epsilon$, where $\omega_m$ are the coefficients and $\epsilon$ is the residue. By enforcing sparsity such that only a small proportion of $\omega_m$ have non-zero values, the value vectors can be interpreted in terms of a few text concepts. More implementation details can be found in the Appendix Sec. A.2.

**Visual Perspective.** For each knowledge neuron, the coefficient is calculated as the inner product between the query vector and the residual stream. For example, in the case of FFN neuron L$l$F$k$, the coefficient for the class token is expressed as $(\mathbf{W}^l_{fc1,k,:})^T(h_0^{l-1} + A_0^l)$. Here, we omit the nonlinear transformation, as it generally does not affect the relative ranking of the coefficients. The residual stream can be reformulated as: $h_0^0 + \sum_{z=1}^{l-1} F_0^z + \sum_{z=1}^l A_0^z$. The contribution of $h_0^0 + \sum_{z=1}^{l-1} F_0^z$ always originates from the class token. However, we observe that the contribution of the third term

can be attributed to individual patches. Specifically, we can further decompose it as:

$$(\mathbf{W}_{fc1,k,:}^l)^T \sum_{z=1}^l A_0^z = \sum_{p=0}^N (\mathbf{W}_{fc1,k,:}^l)^T \sum_{z=1}^l \sum_{j=1}^H \alpha_{0,j,p}^z \mathbf{W}_{o,j}^z \mathbf{W}_{v,j}^z h_p^{z-1}. \quad (6)$$

Therefore, the contribution of this term can be attributed to individual patch. A similar derivation applies for knowledge neurons in the MSA blocks. By calculating this value for each patch and visualizing it as a heatmap, we gain an intuitive understanding of which regions of the input image are important for activating the knowledge neuron. This, in turn, provides a visual explanation of the neuron's behavior. This framework also allows us to allocate contributions to individual neurons in the earlier layers, capturing their second-order effects on the entity recognition. Interestingly, we find for a given knowledge neuron, there are often a few neurons with significant second-order effects. We call them the *query neurons* of the knowledge neuron. More details are in Sec. A.3.

## 4 EXPERIMENTS

### 4.1 EXPERIMENTAL SETTINGS

**Dataset.** To evaluate the effectiveness of the proposed knowledge neuron attribution techniques and analyze the behavior of knowledge neurons in VLMs, we create a dataset called *VisEnt*, specially designed for detecting specific entities. VisEnt includes six categories: 1) celebrities, 2) cartoon characters, 3) commercial logos, 4) city landmarks, 5) artworks, and 6) films. For each category, we select over ten diverse entities and collect multiple images per entity, covering various contexts and perspectives. More details about VisEnt are provided in the Appendix Sec. A.4. Additionally, we compare neuron activation patterns between specific entities and general objects, using sampled images from ImageNet's (Deng et al., 2009) validation set with ImageNet labels as prompts.

**Implementation Details.** Unless stated otherwise, we use CLIP ViT-L-14, pretrained by OpenAI, as the VLM for our study. Experimental results for other VLMs are provided in the Appendix Sec. A.9. All experiments are performed on NVIDIA GeForce RTX 3090 GPUs.

### 4.2 COMPARISON WITH BASELINE METHODS

**Comparison with other neuron-attribution methods.** To show the superiority of our attribution method, we conduct comparisons with three widely-used baseline methods: (1) *Activation-based* method: using the activation coefficient of each neuron as the importance score; (2) *Gradient-based* method: calculating the gradient of the final cosine similarity w.r.t the activation as the importance score; and (3) *Combined* method: using the product of the activation and gradient scores as the importance score. For evaluation, we measured the cosine similarity between the textual representations of the specific entity's name and the textual interpretation of the top attributed neurons. The experiments are conducted on the entire VisEnt dataset. The results in Table 1 demonstrate that our method consistently achieves higher scores compared to these baselines.

Table 1: Comparison of attribution methods.

| Methods | Cosine (%) |
|---|---|
| Activation | 70.13 |
| Gradient | 69.05 |
| Combine | 65.43 |
| Ours | **82.02** |

**Comparison of heatmap techniques.** we conduct deletion experiments to evaluate the importance of the highlighted region of the heatmap. Specifically, we mask the top 10%, 20% and 30% pixels with the highest importance scores. We then do inference with the masked image. We compare the activation coefficient of the corresponding knowledge neuron and the visual-textual cosine similarity to the corresponding labels,

Table 2: Comparison of heatmaps methods.

| Metrics | (%)masked | 0% | 10% | 20% | 30% |
|---|---|---|---|---|---|
| | Random | 4.14 | 3.06 | 2.81 | 2.84 |
| Activation | Act | 4.14 | 2.47 | 1.98 | 1.43 |
| | Ours | 4.14 | **1.94** | **1.45** | **1.02** |
| | Random | 25.18 | 25.32 | 24.93 | 24.65 |
| Cosine | Act | 25.18 | 24.32 | 23.57 | 23.04 |
| | Ours | 25.18 | **23.99** | **22.55** | **21.41** |

before and after the perturbation. We also compare with two baselines: (1) *random masks* and (2) *neuron activation maps*. The results are shown in Table 2. With 30% of the pixels masked, the neuron activation decreases by 75.36%, and the visual-textual cosine similarity decreases by 14.97%. This shows the detected region actually activates the neuron and contributes to the association.

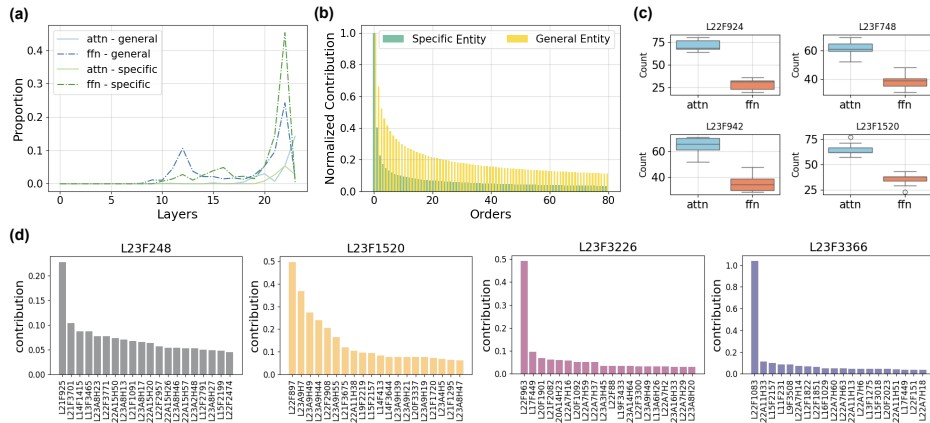

Figure 3: **Behaviors of knowledge neurons.** (a) The distribution of knowledge neurons within ATTN and FFN blocks across various layers. (b) Normalized decay curve of neuronal contribution ranked by significance. (c) Distribution of query neurons for sample knowledge neurons. (d) Decay curves of second-order contributions from query neurons ranked by significance.

### 4.3 RESULTS OF KNOWLEDGE NEURONS ATTRIBUTION.

**Associative knowledge neurons consolidate visual features and trigger the associated memories.** Fig. 2 presents several examples of the detected knowledge neurons, which are associated with well-known specific entities spanning diverse categories. Interpretations from both visual and linguistic perspectives illustrate the strong connection between the knowledge neurons and their corresponding entities. The heatmap effectively identifies the entity within images, even in complex scenes. For instance, the neuron linked to "Taylor Swift" can accurately distinguish her from individuals who share the same gender, race, age, and even hair color. The decomposed language concepts are also closely tied to the entities: they may represent the entity's name (e.g., Starbucks) or something explicitly related to it (e.g., Snoopy is one of the main characters in the comic strip Peanuts). As the decomposition is based on only 20k most commonly used words, the results may not be optimal. However, it still provides valuable insights into the knowledge encoded within the neurons.

**Knowledge neurons are concentrated in the final few FFN layers.** For each entity in VisEnt, we identify the top three neurons that contribute the most to alignment with the corresponding labels. The distribution of these knowledge neurons is depicted in Fig. 3 (a). For comparison, we plot a similar distribution for images sampled from ImageNet, illustrating how neurons activate for general objects. The results reveal that knowledge neurons related to specific entities are predominantly concentrated in the final few FFN layers. In contrast, for general objects, influential neurons may be located in the shallow layers or MSA blocks. A hypothesis is that ViTs process simpler concepts in the shallower layers and progressively integrate them into more complex concepts, including specific entities, in the deeper layers. We will demonstrate the hypothesis in Appendix Sec. A.5.

**A small number of knowledge neurons account for the majority of visual-text alignment.** We further examine how alignment contribution is distributed across neurons. For each sample, we identify the 80 neurons with the highest contributions and rank them by contribution magnitude. We then compute the average contribution for each rank position across all samples. The decay curve, illustrating how neuronal contribution decreases with rank, is shown in Fig. 3 (b). It has been normalized so that the contribution of top neuron is 1. The plots are presented for both VisEnt and ImageNet. Compared to general objects, the contribution for specific entities diminishes much more rapidly, with only a small number of neurons playing a crucial role in alignment. This suggests that the visual representations of general objects may result from the combined influence of many neurons, whereas specific entities are primarily defined by a small set of associative knowledge neurons.

**Knowledge neurons are associated with relatively fixed query neurons.** In Fig. 3 (c), we present the distribution of the top 10 query neurons for several knowledge neurons. A consistent finding is that, unlike knowledge neurons, most query neurons gather at the MSA block. This is expected as MSA blocks are responsible for transferring information from patch tokens to the class token, with

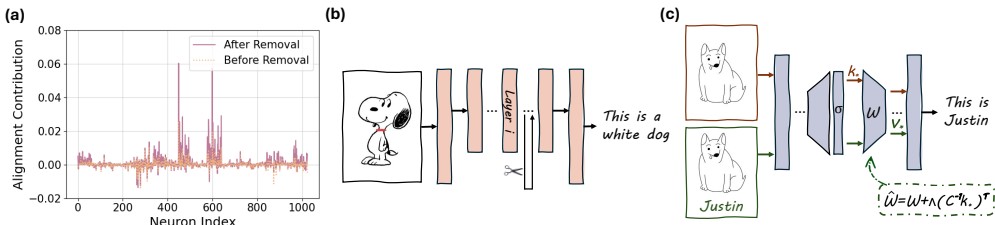

Figure 4: **Knowledge editing.** (a) Removing knowledge neurons increases alignment contribution of neurons in subsequent layers. (b) Knowledge removal by suppressing knowledge neuron contributions in the residual stream. (c) ROME knowledge insertion guided by captioned images.

Table 3: **Results of knowledge removal.** We report name-image similarity (%) before and after editing the top knowledge neurons (KNs). We include similarity with irrelevant images as a lower bound and post-edit ImageNet zero-shot accuracy for reference. Each case refers to a celebrity.

| cosine(%) | case 1 | case 2 | case 3 | case 4 | case 5 | case 6 | case 7 | case 8 | case 9 | case 10 | average |
|---|---|---|---|---|---|---|---|---|---|---|---|
| original | 22.36 | 22.94 | 25.46 | 21.54 | 24.12 | 24.59 | 19.45 | 20.85 | 24.63 | 23.28 | 22.92 |
| − top-1 KN | 14.84 | 17.99 | 22.77 | 17.16 | 18.71 | 17.65 | 15.09 | 16.09 | 21.81 | 18.12 | 18.02 |
| − top-3 KNs | 12.63 | 16.90 | 21.71 | 15.16 | 17.20 | 15.98 | 14.19 | 15.42 | 19.91 | 17.00 | 16.61 |
| lower bound (mean) | 13.33 | 14.42 | 14.93 | 13.13 | 10.82 | 11.46 | 14.83 | 12.88 | 15.81 | 14.08 | 13.57 |
| lower bound (max) | 15.62 | 17.17 | 19.52 | 15.23 | 16.51 | 13.52 | 17.10 | 17.27 | 21.02 | 16.96 | 16.99 |
| ImageNet acc | 72.22 | 72.20 | 72.20 | 72.20 | 72.26 | 72.16 | 72.20 | 72.20 | 72.18 | 72.20 | 72.20 |

the query neurons interacting with the patch tokens and then activating the knowledge neurons. In Fig. 3 (d), we display the average second-order effects of the top query neurons for several knowledge neurons, averaged across samples. We observe that the second-order effects mainly come from a few query neurons. This shows that the query neurons remain fixed for a given knowledge neuron. Moreover, as standard neurons, the query neurons can be further interpreted, and are attached with other query neurons. Some query neurons also function as knowledge neurons for other concepts. This enables us to trace how general concepts are progressively composed into specific entities. And we notice in the process, the same set of knowledge and query neurons are responsible for recognizing entities represented by both *photograph* and *caption words*. We show detailed examples in Sec. A.5.

**Knowledge neurons suppress the activations of other neurons.** While knowledge neurons play a crucial role in recognizing specific entities, we find that directly removing a knowledge neuron does not eliminate the knowledge associated with the entity. The cosine similarity remains nearly unchanged before and after the neuron's removal. To investigate this phenomenon, we select a knowledge neuron from the FFN layer and visualize the alignment contribution (towards the associated entity) of neurons in the subsequent attention layer, both with and without the knowledge neuron. As shown in Fig. 4 (a), we observe that after the knowledge neuron is removed, the alignment contribution of most neurons increases. This suggests a hypothesis that knowledge neurons inhibit the activation of other neurons. Removing a knowledge neuron allows these suppressed neurons to respond more strongly to the specific entity, ultimately maintaining similar similarity scores before and after the removal. A more detailed analysis of this phenomenon is provided in Appendix Sec. A.6.

### 4.4 APPLICATIONS IN KNOWLEDGE EDITING

**Removal of Sensitive Knowledge.** Directly deleting knowledge neurons fails to unlearn the associated facts. Instead, suppressing their residual-stream contributions, a standard steering technique, reduces the knowledge (Fig. 4 (b)). We test this on celebrity entities to mitigate sensitive personal information. For each celebrity, we locate the relevant knowledge neurons, apply the suppression edits, and measure image–name similarity. We also evaluate zero-shot ImageNet accuracy to gauge generalization impact. As shown in Table 3, suppressing the top three neurons lowers cosine similarity by 6.31% with minimal zero-shot degradation. As lower bounds, we use photos of unrelated individuals matched by race and gender to the target and compute their name–image similarity. We report both mean and max baseline similarity. Although the post-edit similarity remains higher than the mean baseline similarity, it aligns the maximum baseline similarity. This shows that after

Table 4: **Results of knowledge insertion.** Each case corresponds to a specific type of flower or bird. We report the zero-shot accuracy (%) for the target category, the full dataset, and ImageNet.

| methods | accuracy | Flower-102 | | | | | | CUB-200 | | | | average |
| | | case 1 | case 2 | case 3 | case 4 | case 5 | case 6 | case 7 | case 8 | case 9 | case 10 | |
|---|---|---|---|---|---|---|---|---|---|---|---|---|
| **original** | category | 0.00 | 0.00 | 0.00 | 0.00 | 2.56 | 3.85 | 3.33 | 0.00 | 0.00 | 0.00 | 0.97 |
| | dataset | 79.50 | 79.50 | 79.50 | 79.50 | 79.50 | 62.10 | 62.10 | 62.10 | 62.10 | 62.10 | 70.80 |
| | ImageNet | 72.20 | 72.20 | 72.20 | 72.20 | 72.20 | 72.20 | 72.20 | 72.20 | 72.20 | 72.20 | 72.20 |
| **finetuning** | category | 78.57 | 100.00 | 93.75 | 83.33 | 94.87 | 84.62 | 93.33 | 73.33 | 53.33 | 96.67 | 85.18 |
| | dataset | 74.67 | 77.43 | 75.60 | 73.90 | 74.42 | 55.71 | 58.78 | 56.45 | 54.14 | 58.18 | 65.93 |
| | ImageNet | 71.72 | 71.36 | 71.34 | 72.20 | 71.98 | 72.04 | 72.14 | 72.10 | 72.06 | 72.02 | 71.90 |
| **LoRA** | category | 78.57 | 100.00 | 93.75 | 83.33 | 94.87 | 92.31 | 93.33 | 76.67 | 50.00 | 96.67 | 85.95 |
| | dataset | 75.72 | 78.23 | 78.56 | 78.77 | 73.57 | 55.66 | 57.92 | 48.41 | 50.67 | 55.35 | 65.29 |
| | ImageNet | 72.06 | 72.24 | 72.38 | 72.12 | 72.22 | 72.14 | 72.12 | 71.94 | 71.92 | 72.10 | 72.12 |
| **knowledge insertion (ours)** | category | 78.57 | 100.00 | 96.88 | 83.33 | 94.87 | 92.31 | 93.33 | 80.00 | 50.00 | 96.67 | 86.60 |
| | dataset | 80.51 | 79.21 | 80.63 | 80.71 | 79.70 | 61.74 | 62.29 | 61.39 | 61.84 | 61.25 | 70.93 |
| | ImageNet | 72.24 | 72.22 | 72.18 | 72.12 | 72.18 | 72.12 | 72.34 | 72.24 | 72.34 | 72.36 | 72.23 |

knowledge removal, the average image-name similarity for the target entity generally drops below that of the irrelevant (but somewhat similar) individuals. Therefore, the method has practical value, as reducing the model's sensitivity to private information is often sufficient in privacy protection tasks.

**Insertion of New Knowledge.** The analysis in Sec. 4.3 highlights two key findings: 1) the recognition of specific entities is primarily driven by a small number of neurons, and 2) knowledge neurons respond to both graphical signals and caption signals within an image. Rule 1 allows us to leverage low-rank model editing to insert new knowledge, while rule 2 suggests using captioned images as guidance of new knowledge. Based on these insights, we propose a knowledge insertion method based on RMOE (Meng et al., 2022a) to teach VLMs to recognize previously unknown entities (see Fig. 4 (c)). ROME edits the weight (i.e., $\mathbf{W}_{fc2}^l$ and $\mathbf{W}_o^l$) by solving the following problem:

$$\min \|\hat{W}K - V\|_F \quad \text{s.t.} \ \hat{W}k_* = v_* \quad \text{by setting} \ \hat{W} = W + \Lambda(C^{-1}k_*)^T. \tag{7}$$

Here, we define each $(k, v)$ pair as the input and output of the targeted layer (e.g., $k = \sigma(\mathbf{W}_{fc1}^l(h_i^{l-1} + A_i^l))$ and $v = F_i^l$ for editing $\mathbf{W}_{fc2}^l$). $K$ and $V$ represent sets of pre-cached vector keys and values, where $K = [k_1|k_2|\cdots]$ and $V = [v_1|v_2|\cdots]$, calculated from images sampled from ImageNet. $C = KK^T$ is a constant, and $\Lambda = (v_* - Wk_*)/(C^{-1}k_*)^T k_*$ is a vector proportional to the residual error of the key-value pair for the target entities. We notice that CLIP can recognize an object when its name is overlaid on the image. Based on it, we use the input corresponding to original images as $k_*$ and the output corresponding to captioned images as $v_*$. These vectors are calculated as the average across several images ($\sim$20) to improve robustness. By doing so, we obtain the same output as captioned images when the input image is caption-free. For implementation details, see Sec. A.7. We evaluate knowledge insertion on CUB-200 and Flower-102. For each dataset, we pick classes with near-zero zero-shot accuracy, insert knowledge for each target class, and measure test accuracy on the target class, the full dataset, and ImageNet. We compare against finetuning the last transformer block and LoRA (Hu et al., 2022), both trained to maximize image–name cosine similarity. As Table 4 shows, our method greatly boosts target-class accuracy with minimal impact on other classes or zero-shot performance. Finetuning can also recognize the new entities but overfits on limited samples, severely harming generalization and making it impractical. In Appendix Sec. A.8, we show the results of scalable knowledge insertion, using the framework of MEMIT (Meng et al., 2022b).

## 5 LIMITATION AND CONCLUSION

In this paper, we investigate the phenomenon of associative knowledge neurons in VLMs. We introduce a framework to localize and interpret these neurons. Based on our findings, we outline several key properties of knowledge neurons and propose two applications for knowledge editing.

This work has certain limitations. First, we simplify by only considering the first-order effects of neurons. Additionally, our experiments are conducted on relatively small datasets. Finally, our study focuses on the behavior of CLIP-style VLMs, with intermediate embedding layers linking modalities.

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

# A APPENDIX

## A.1 LAYER NORMALIZATION

In this section, we outline the modifications required to incorporate layer normalizations into our method. Layer normalizations are applied in two key locations: before the projection layer (on the output of the ViT) and before each layer in the ViT (on the layer input). We detail how the neuron-level contributions should be adjusted accordingly.

**Pre-projection Layer Normalization.** In many implementations of CLIP, a layer normalization LN is applied to the ViT's output before the projection layer. Formally, the image representation for an image $I$ is given by:

$$M_{\text{image}}(I) = \mathbf{PLN}(h_0^L). \tag{8}$$

The normalization layer can be written as:

$$\text{LN}(x) = \gamma * \frac{x - \mu_l}{\sqrt{\sigma_l^2 + \zeta}} + \beta, \tag{9}$$

where $x \in \mathbb{R}^d$ is the input token, $\mu_l, \sigma_l \in \mathbb{R}$ are the mean and standard deviation, and $\gamma, \beta \in \mathbb{R}^d$ are learned vectors. To incorporate layer normalization into our framework, we must adjust the projection process used to map the intermediate output or value vectors of the neurons into the shared representation space. Specifically, when calculating the intermediate representation (e.g., $h_0^{l-1}$) and the neuron value (e.g., $\mathbf{W}_{fc2,:,k}^l$), we apply layer normalization. This involves first computing the mean $\hat{\mu}$ and standard deviation $\hat{\sigma}$ from the vectors (e.g., $\hat{\mu} = \text{mean}(h_0^{l-1})$), followed by performing a linear transformation using the parameters $\gamma$ and $\beta$.

**MSAs and FFNs Input Layer Normalizations.** In the main paper, we do not describe the normalization layers that are applied to each input of FFN and MSA block in the model. More accurately, the complete representation of MSA block and FFN block are:

$$A_i^l = \text{ATTN}_i^l(\text{LN}_a^l(h_0^{l-1}), \cdots, \text{LN}_a^l(h_N^{l-1})), \quad F_i^l = \text{FFN}(\text{LN}_f^l(A_i^l + h_i^{l-1})), \tag{10}$$

where $\text{LN}_a^l$ and $\text{LN}_f^l$ are the layer normalizations applied to each token in the input matrix of the MSA blocks and FNN layers. This modification does not impact our corollaries regarding the direct contributions of the FFN layers and MSA blocks, as it solely pertains to the outputs of these layers. Specifically, we only need to modify Eq. 4 and Eq. 5 as:

$$\text{Imp}(\text{L}l\text{F}k) = \cos(\mathbf{P}(\text{LN}_f^l(h_0^{l-1} + A_0^l) + m_F^l(k)\mathbf{W}_{fc2,:,k}^l), M_{\text{text}}(T)) - \cos(\mathbf{PLN}_f^l(h_0^{l-1} + A_0^l), M_{\text{text}}(T)), \tag{11}$$

where $m_F^l(k) = \sigma((\mathbf{W}_{fc1,k,:}^l)^T \text{LN}_f^l(h_0^{l-1} + A_0^l))$, and:

$$\text{Imp}(\text{L}l\text{A}j\text{H}k) = \cos(\mathbf{P}(\text{LN}_a^l(h_0^{l-1}) + m_A^l(j,k)\mathbf{W}_{o,j,:,k}^l), M_{\text{text}}(T)) - \cos(\mathbf{PLN}_a^l(h_0^{l-1}), M_{\text{text}}(T)), \tag{12}$$

where $m_A^l(j,k) = \sum_{p=0}^N \alpha_{0,j,p}^l (\mathbf{W}_{v,j,k,:}^l)^T \text{LN}_a^l(h_p^{l-1})$.

## A.2 IMPLEMENTATION DETAILS FOR LINGUISTIC NEURON INTERPRETATION

To interpret the knowledge neuron from a linguistic perspective, we represent its value vector as a weighted sum of concept representations. For this purpose, we construct a concept dictionary using the 20,000 most frequently used English words (Source), which has been widely utilized in prior studies (Oikarinen & Weng, 2023b; Rao et al., 2024). For sparse decomposition, we leverage the orthogonal matching pursuit implementation from scikit-learn library, setting the hyperparameter for the number of non-zero coefficients to 5. To ensure all value vectors contribute positively to alignment, we normalize them by multiplying both the query vector and the value vector by $-1$ if the neuron's coefficient (e.g., $m_F^l(k)$ or $m_A^l(j,k)$, averaged across all images of the corresponding specific entities) is negative while its alignment contribution is positive. This normalization preserves the network's output while ensuring that the contributions are directionally consistent.

### A.3 SECOND-ORDER EFFECTS OF QUERY NEURONS

In the main text, we explain how to decompose the alignment contribution of a knowledge neuron into the second-order effects of patch tokens within the MSA blocks and generate corresponding saliency maps as visual explanations for the knowledge neurons. We observe that attribution can be performed at a more fine-grained level, specifically for each neuron in the previous layer. Neurons with the highest second-order contributions are referred to as the query neurons of the knowledge neuron. Specifically, we consider the following two cases.

**Knowledge Neuron from the FFN Layers.** Given an FFN neuron L$l$F$k$, the coefficient for the class token is expressed as:

$$(\mathbf{W}_{fc1,k,:}^l)^T(h_0^{l-1} + A_0^l) = (\mathbf{W}_{fc1,k,:}^l)^T(h_0^0 + \sum_{z=1}^{l-1} F_0^z + \sum_{z=1}^{l} A_0^z) \tag{13}$$

$$=(\mathbf{W}_{fc1,k,:}^l)^T(h_0^0 + \sum_{z=1}^{l-1} \mathbf{W}_{fc2}^z \sigma(\mathbf{W}_{fc1}^z(h_i^{z-1} + A_i^z)) + \sum_{p=0}^{N}\sum_{z=1}^{l}\sum_{j=1}^{H} \alpha_{0,j,p}^z \mathbf{W}_{o,j}^z \mathbf{W}_{v,j}^z h_p^{z-1}). \tag{14}$$

We trace the contribution of each neurons to this term. For an FFN neuron L$l'$F$k'$, the contribution is:

$$\mathcal{E}(Ll'Fk' \to LlFk) = \sigma((\mathbf{W}_{fc1,k',:}^{l'})^T(h_i^{l'-1} + A_i^{l'}))(\mathbf{W}_{fc1,k,:}^l)^T \mathbf{W}_{fc2,:,k'}^{l'}. \tag{15}$$

Similarly, for a neuron from the MSA block, L$l'$A$j'$H$k'$, the contribution term is:

$$\mathcal{E}(Ll'Aj'Hk' \to LlFk) = \sum_{p=0}^{N} \alpha_{0,j',p}^{l'} (\mathbf{W}_{v,j',k,:}^{l'})^T h_p^{l'-1}(\mathbf{W}_{fc1,k',:}^l)^T \mathbf{W}_{o,j',:,k'}^{l'}. \tag{16}$$

**Knowledge Neuron from the MSA blocks.** Given an MSA neuron L$l$A$j$H$k$, the coefficient for the class token is expressed as:

$$(\mathbf{W}_{v,j,k,:}^l)^T \sum_{p=0}^{N} \alpha_{0,j,p}^l h_p^{l-1} = (\mathbf{W}_{v,j,k,:}^l)^T \sum_{p=0}^{N} \alpha_{0,j,p}^l(h_p^0 + \sum_{z=1}^{l-1} F_p^z + \sum_{z=1}^{l-1} A_p^z) \tag{17}$$

$$=(\mathbf{W}_{v,j,k,:}^l)^T \sum_{p=0}^{N} \alpha_{0,j,p}^l(h_p^0 + \sum_{z=1}^{l-1} \mathbf{W}_{fc2}^z \sigma(\mathbf{W}_{fc1}^z(h_p^{z-1} + A_p^z)) + \sum_{z=1}^{l-1}\sum_{j=1}^{H}\sum_{p'=0}^{N} \alpha_{p,j,p'}^z \mathbf{W}_{o,j}^z \mathbf{W}_{v,j}^z h_{p'}^{z-1}). \tag{18}$$

To this end, the second-order contribution of an FFN neuron L$l'$F$k'$ is:

$$\mathcal{E}(Ll'Fk' \to LlAjHk) = \sum_{p=0}^{N} \alpha_{0,j,p}^l \sigma((\mathbf{W}_{fc1,k',:}^{l'})^T(h_p^{l'-1} + A_p^{l'}))(\mathbf{W}_{v,j,k,:}^l)^T \mathbf{W}_{fc2,:,k'}^{l'}. \tag{19}$$

Similarly, the second-order effects of an MSA neuron L$l'$A$j'$H$k'$ is:

$$\mathcal{E}(Ll'Aj'Hk' \to LlAjHk) = \sum_{p=0}^{N}\sum_{p'=0}^{N} \alpha_{0,j,p}^l \alpha_{p,j',p'}^{l'} (\mathbf{W}_{v,j',k',:}^{l'})^T h_{p'}^{l'-1}(\mathbf{W}_{v,j,k,:}^l)^T \mathbf{W}_{o,j',:,k'}^{l'}. \tag{20}$$

### A.4 DETAILED DESCRIPTION OF *VisEnt* DATASET

Here, we provide a detailed description of the *VisEnt* dataset. The dataset comprises images categorized into six specific groups of entities:

- *Celebrities*: This category includes notable public figures such as musicians, and athletes.
- *Cartoon Characters*: Featuring iconic characters from popular animations and franchises.

Table 5: List of specific entities included in the *VisEnt* dataset.

| category | specific entities |
|---|---|
| Celebrities | Barack Obama, Beyonce, Bruce Lee, Donald Trump, Hillary Clinton, JK Rowling, Lady Gaga, Michael Jordan, Steve Jobs, Taylor Swift |
| Cartoon characters | Cinderella, Mario, Mickey, Pikachu, Shrek, Simpson, Snoopy, Spiderman, Superman |
| Commercial logos | Apple, Chanel, Coca-Cola, Fedex, Mercedes-Benz, NBC, NIKE, Starbucks, Toyota |
| City & landmarks | Beijing, Grand Canyon, Great Barrier Reef, the Great Wall, Mount Everest, Mumbai, New York City, Niagara Fall, Sydney Opera House, Taj Mahal |
| Artworks | David, Discobolus, Girl with a pearl earring, Guernica, Last supper, Mona Lisa, Scream, Starry night, Sunflowers, Thinker |
| Films | Forrest Gump, Friends, Harry Potter, Roman Holiday, Schindler's List, Star Wars, The Godfather, The Lord of the Rings, Titanic, Twilight |

- *Commercial Logos*: A collection of logos from well-known global brands across industries.
- *City Landmarks*: Images of prominent landmarks that showcase cultural significance.
- *Artworks*: Renowned paintings and sculptures from different time periods and styles.
- *Films*: Includes iconic visuals and posters from popular movies across genres.

For each category, we carefully selected 10 representative entities, ensuring a diverse representation in terms of factors such as demographics, geographic location, and industry. For instance, celebrities were chosen to include individuals from different professions and regions, while city landmarks covered a mix of historical and modern sites from various countries. For each selected entity, we gathered over 10 images from the internet, ensuring diversity in viewpoints, backgrounds, and contexts. It highlights variations in appearance and provides a comprehensive visual representation of each entity. The complete list of specific entities included in the dataset is presented in Table 5.

### A.5 VISUALIZATION FOR CONCEPT EVOLUTION

Since query neurons are also standard neurons, they can be further interpreted, and their associated query neurons can be identified. Interestingly, some query neurons also function as knowledge neurons for other concepts. This enables us to trace how general concepts are progressively composed into specific entities. In Fig. 5, we show several of the examples. These figures are created by first identifying the knowledge neurons and their associated query neurons. Next, we apply neuron dissection techniques to interpret the query neurons. Subsequently, we treat the query neuron as a knowledge neuron and repeat the process. This approach allows us to trace back how visual information is progressively processed by neurons across different layers.

For example, in Fig. 5 (a), we show such a revolution path. A clear pathway reveals that the "Taylor" neuron is strongly activated by the "concerts" neuron, which, in turn, is activated by the neuron linked to the concept of "microphone". In Fig. 5 (b), we provide another example where the image contains the caption "Taylor Swift" instead of the photograph. We observe that the same knowledge neuron contributes to the alignment, with its direct query neurons overlapping with those from the previous example. However, the pathway diverges in the middle to shallow layers, allowing for the extraction of different visual patterns. This indicates that knowledge neurons can respond to specific entities in various forms, but the activated neurons and their associated query neurons remain consistent.

It is worth noting that not all neurons have clear, human-understandable meanings, although a significant portion of them do. We only showcase neurons associated with meaningful concepts. The role of other neurons remains an open question for further investigation.

### A.6 FURTHER ANALYSIS OF NEURON SUPPRESSION

In Sec. 4.3 of the main text, we identified the phenomenon of neuron suppression. Due to space constraints, we provide a more detailed analysis in this section.

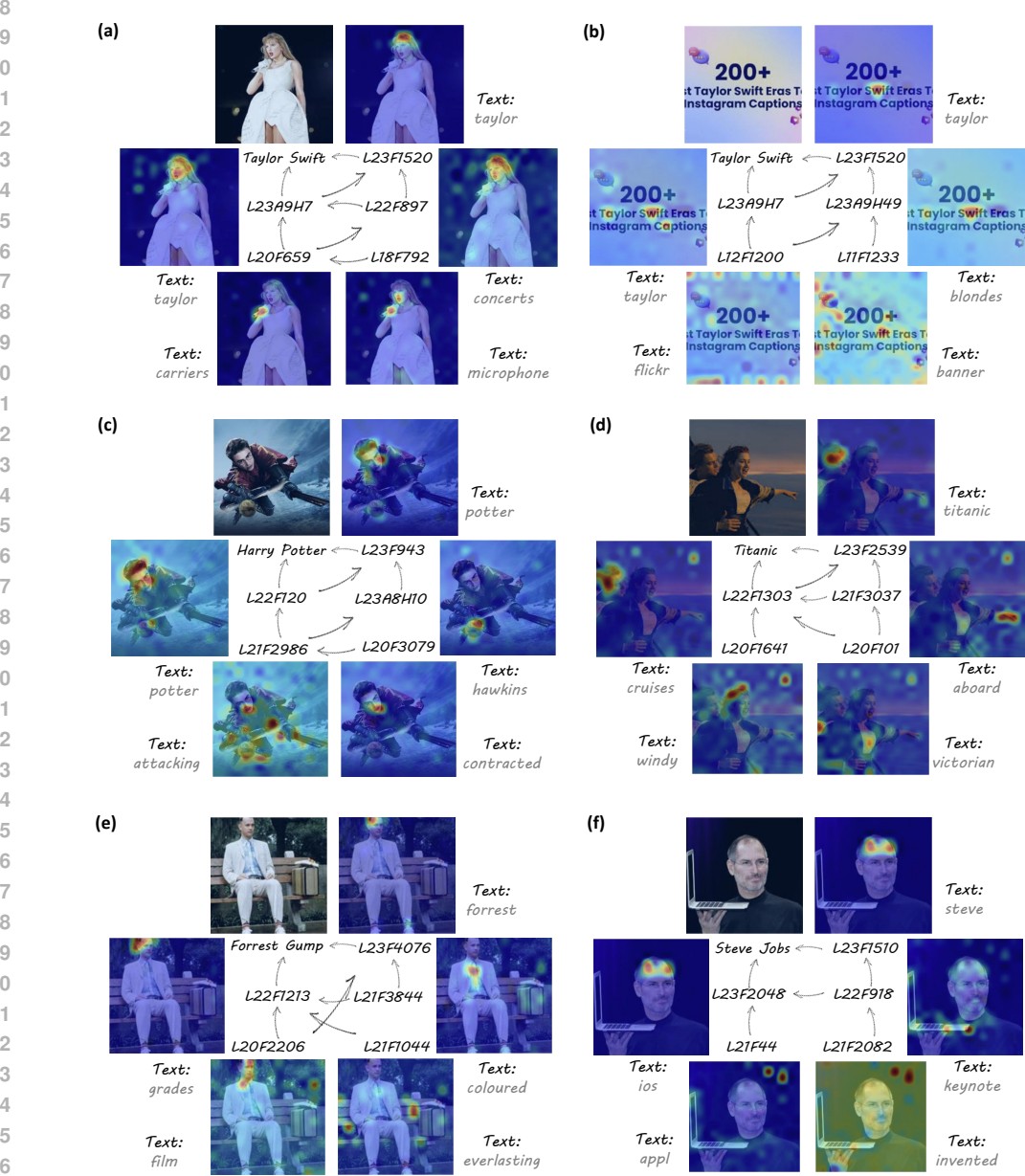

Figure 5: **Examples of concept evolution**. we trace how general concepts are gradually processed into more complex, specific entity concepts by analyzing the interactions between knowledge neurons and their associated query neurons.

As shown in Fig. 4(a), removing a knowledge neuron in an FFN layer by setting its value vector to zero leads to an increase in the alignment contribution of neurons in subsequent MSA blocks. This effect can be attributed to two possible reasons: (1) the reduced base alignment caused by the removal of the knowledge neuron, or (2) the enhanced alignment contribution of subsequent neurons due to the elimination of the suppression effect. To validate the hypothesis that the latter contributes significantly to this increase, we fix the base latent vector and compute the alignment score improvement for neurons in subsequent layers. Formally, we define the improvement of alignment scores contributed by neurons in MSA blocks as:

$$\text{Imp}(\text{L}l\text{A}j\text{H}k) = \cos(\mathbf{P}(h_0^{l-1} + m_A^l(j,k)\mathbf{W}_{o,j,:,k}^l), M_{\text{text}}(T)) - \cos(\mathbf{P}h_0^{l-1}, M_{\text{text}}(T)), \quad (21)$$

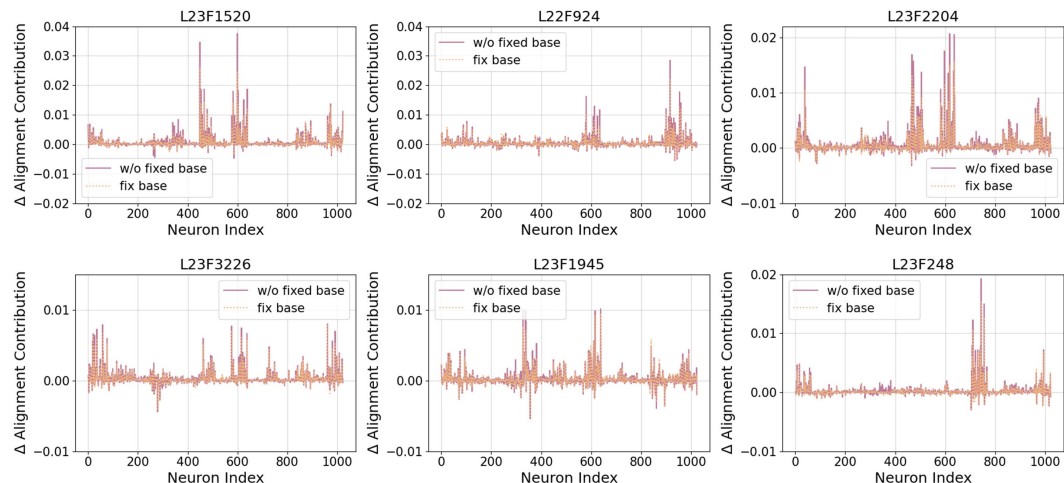

Figure 6: **Illustration of neuron suppression.** We present examples of several knowledge neurons and demonstrate how their removal in the FFN layer impacts alignment. The plots display the resulting increase in alignment contribution of the neurons in subsequent MSA blocks.

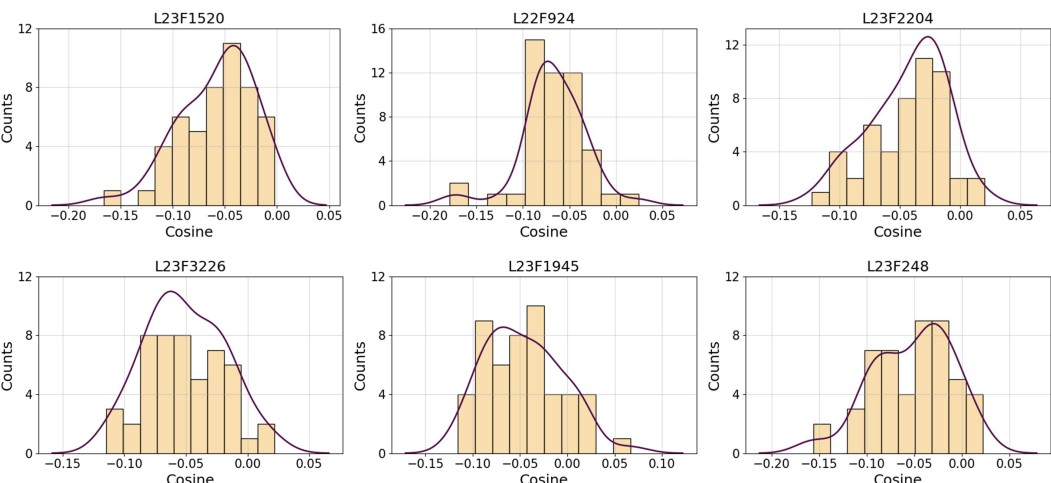

Figure 7: **Illustration of neuron suppression.** Distribution of the cosine similarity scores between the value vector of the knowledge neurons and query vectors of the top 50 suppressed neurons.

where the base latent vector $h_0^{l-1}$ remains unchanged (i.e., not removing the knowledge neurons). We then calculate the change in alignment improvement, $\Delta\text{Imp}(\text{L}l\text{A}j\text{H}k)$, after removing the knowledge neuron. Under such fixed base vector setting, the change is primarily driven by $m_A^l(j,k)\mathbf{W}_{o,j,:,k}^l$. Results in Fig. 6 show that even with a fixed base latent vector, there is a clear increasing trend in the alignment contribution of most neurons in subsequent layers. Furthermore, these improvements are close to the full observed improvements, supporting the conclusion that knowledge neurons suppress the activation of other neurons associated with the target entities.

To further explain this suppression phenomenon, we analyze it from the perspective of second-order effects between neurons. As shown in Eq. 20, the second-order effect of an FFN neuron on an MSA neuron is proportional to the inner product between their value vector and query vector, specifically $(\mathbf{W}_{v,j,k,:}^l)^T\mathbf{W}_{o,j',:,k'}^{l'}$. Suppression primarily arises when these vectors point in opposite directions, i.e., when $\mathbf{W}_{v,j,k,:}^l$ and $\mathbf{W}_{o,j',:,k'}^{l'}$ have a negative cosine similarity. In Fig. 7, we examine several knowledge neurons. For each, we select the top 50 neurons in subsequent MSA blocks that experience the greatest suppression effect, measured as the highest increase in $\text{Imp}(\text{L}l'\text{A}j'\text{H}k')$ after the knowledge neuron $\text{L}l\text{F}k$ is removed. To ensure all value vectors contribute positively to alignment,

Table 6: Comparison of visual-textual similarity decrease of the models after complete neuron removal and neuron-effects suprresion in the residue stream.

| Models | Original cosine | Complete neuron removal | Residual suppression |
|---|---|---|---|
| CLIP | 23.71 | 23.76 | 21.48 |
| DFN | 33.06 | 32.94 | 31.26 |
| MetaCLIP | 28.62 | 28.50 | 26.72 |

Table 7: The mapping table from Index in Table 3 and Table 4 to the real names of the entities.

| Index | Table 3 | Table 4 |
|---|---|---|
| case 1 | Taylor Swift | ball moss |
| case 2 | Lady Gaga | bishop of llandaff |
| case 3 | Bruce Lee | cape flower |
| case 4 | Hillary Clinton | prince of wales feathers |
| case 5 | Steve Jobs | sword lily |
| case 6 | JK Rowling | chuck will widow |
| case 7 | Donald Trump | geococcyx |
| case 8 | Barack Obama | nelson sharp tailed sparrow |
| case 9 | Michael Jordan | sayornis |
| case 10 | Beyonce | western gull |

we multiply both $\mathbf{W}^{l'}_{o,j',:,k'}$ and $\mathbf{W}^{l'}_{v,j',k',:}$ by $-1$ if the initial coefficient of the neuron $Ll'Aj'Hk'$ is negative while its alignment contribution is positive. This normalization keeps the outputs of the network unchanged while normalizing the contributions. We then compute the cosine similarity between $\mathbf{W}^{l}_{v,j,k,:}$ and $\mathbf{W}^{l'}_{o,j',:,k'}$ and visualize the distribution as histograms in Fig. 7. The results reveal that most cosine similarities are negative, indicating that the inclusion of knowledge neurons reduces the activation coefficients of positively contributing neurons in subsequent layers, thereby suppressing their contributions. This further proves the neuron suppression hypothesis.

In addition, we conducted a scaled experiment to examine the universality of the neuron suppression phenomenon. Specifically, we sampled 5k images from ImageNet and used the text representation corresponding to each class label to identify the neuron that contributes most to the final visual representation. We then performed two interventions: (1) complete removal of the neuron, and (2) remove the neuron's effect in the residual stream. We compared the visual-textual cosine similarity of the representations before and after each intervention. We conduct experiments on three CLIP-style models: 1) CLIP, 2) DFN (Fang et al., 2024) and 3) MetaCLIP (Xu et al., 2024). The results are shown in Table 6, which consistently showed that the representations remained highly similar after neuron removal. This indicates that neuron suppression is not an isolated artifact of specific entities or architectures but rather a prevalent phenomenon.

This suppression phenomenon reflects a form of competitive inhibition, where dominant knowledge neurons limit the contributions of other neurons. Removing these neurons reduces their influence, enabling others to contribute more freely. Such behavior is consistent with the competitive dynamics observed in biological neural systems and supports the model's ability to allocate representational resources effectively. As future work, we aim to further investigate the underlying reasons for this behavior and its role in the model's representation learning.

## A.7 IMPLEMENTATION DETAILS FOR KNOWLEDGE EDITING

In this section, we provide a more detailed explanation of the setups and implementation specifics for the knowledge editing experiments.

**Knowledge Removal.** We begin by using the neuron attribution algorithm to identify the knowledge neurons that play a significant role in recognizing the entity. For knowledge removal, we target and eliminate their first-order effects, which directly influence the final visual representation via the residual stream. However, the input to the subsequent layer continues to carry the contributions of the

knowledge neurons. As previously analyzed, this approach preserves the suppression effects of the knowledge neurons, mitigating the response from the neurons in subsequent layers to the entity.

To evaluate the performance of the knowledge removal algorithm, we use the celebrity category from the VisEnt dataset, which contains ten celebrities. In each evaluation trial, we remove the information associated with one specific individual and measure the visual-text similarities before and after applying the knowledge removal process. For comparison, we collect photographs of unrelated individuals who share the same race and gender as the targeted celebrity. We also compute the representation similarities between these unrelated individuals' photographs and the name of the targeted celebrity, as encoded by the model after the knowledge removal. Table 8 provides the names of the celebrities corresponding to the cases presented in Table 3.

**Knowledge Insertion.** For knowledge insertion, we use training data from CUB-200 (Wah et al., 2011) or Flower-102 (Nilsback & Zisserman, 2008), apply the ROME algorithm, and evaluate the model's performance on the test set in terms of its ability to recognize the targeted categories. To harness the multi-modal capability of knowledge neurons, we use captioned images as a bridge. Specifically, for the targeted category, we collect images from the training set and overlay each photograph with a caption that names the entity. Examples of these captioned images are shown in Fig. 8. CLIP can associate the captioned images with their corresponding text labels. Next, we apply the ROME algorithm to update the network's weights, ensuring that the output for the original image aligns closely with the output of the captioned image. As a result, during inference, the model can recognize the object even when the image lacks a caption. The key steps of the ROME algorithm are as follows:

- *Determining the Layer to Edit:* We utilize causal tracing (Meng et al., 2022a; Palit et al., 2023) to identify the most effective layer for editing. Specifically, we first pre-cache the outputs of each individual FFN and MSA block when the captioned image is used as input. We then switch to using the original image as input and intervene by replacing the output of each block with the corresponding output from the captioned image. This intervention is performed block by block, and we record the increase in the final visual-text similarity. The layer that produces the greatest improvement is selected as the target layer for editing. Notably, in most cases, the selected layer is one of the final few FFN or MSA layers, which aligns closely with the distribution of knowledge neurons shown in Fig. 3.

- *Choosing $k_*$ to Select the Visual Patterns:* The vector $k_*$ is chosen as the input to the targeted layer when the model is provided with the original images as inputs. If the targeted layer is a FFN layer, the input is

$$k_* = \sigma(\mathbf{W}_{fc1}^l(h_i^{l-1} + A_i^l)). \tag{22}$$

If the targeted layer is an MSA block, the input is then

$$k_* = \sum_{j=1}^{H} \sum_{p=0}^{N} \alpha_{i,j,p}^l \mathbf{W}_{v,j}^l h_p^{l-1}. \tag{23}$$

To improve robustness, the final $k_*$ is computed as the average across all images (approximately 20) from the training set.

- *Choosing $v_*$ to Update the Associative Memory:* The vector $v_*$ is calculated as the output of the targeted layer when the captioned images are used as model inputs. Similar to $k_*$, it is averaged over all images in the training set.

- *Inserting the Associative Knowledge:* Once we have computed the pair $(k_*, v_*)$, we apply Eq. 25, updating the weights $\mathbf{W}_{fc2}^l$ (for FFN layer) or $\mathbf{W}_o^l$ (for MSA block) with a rank-one update that inserts the new key-value association directly.

Table 4 in the main text presents the experimental results for knowledge insertion, while Table 8 lists the specific names of the entities corresponding to each case in Table 4. It is important to note that, in these experiments, we insert only one piece of associative knowledge at a time, as our primary focus is on studying the behavior and characteristics of knowledge neurons. This approach allows us to better understand their role and functionality, serving as a foundation for exploring potential applications. While we demonstrate knowledge editing as a practical example, the development of a scalable method for multiple simultaneous edits is left for future research.

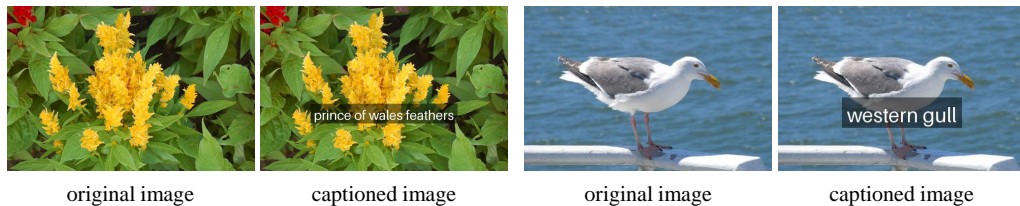

| original image | captioned image | original image | captioned image |

Figure 8: Examples of images with the category names overlaid.

Table 8: **Results of scalable knowledge insertion.** Each case corresponds to a specific type of flower or bird. We report the zero-shot accuracy (%) for the target category, the full dataset, and ImageNet.

| methods | accuracy | Flower-102 | | | | | | CUB-200 | | | | average |
| | | case 1 | case 2 | case 3 | case 4 | case 5 | case 6 | case 7 | case 8 | case 9 | case 10 | |
|---|---|---|---|---|---|---|---|---|---|---|---|---|
| **original** | category | 0.00 | 0.00 | 0.00 | 0.00 | 2.56 | 3.85 | 3.33 | 0.00 | 0.00 | 0.00 | 0.97 |
| | dataset | 79.50 | 79.50 | 79.50 | 79.50 | 79.50 | 62.10 | 62.10 | 62.10 | 62.10 | 62.10 | 70.80 |
| | ImageNet | 72.20 | 72.20 | 72.20 | 72.20 | 72.20 | 72.20 | 72.20 | 72.20 | 72.20 | 72.20 | 72.20 |
| **knowledge insertion (ours)** | category | 71.43 | 96.97 | 96.88 | 83.33 | 92.31 | 92.31 | 90.00 | 70.00 | 33.33 | 96.67 | 80.09 |
| | dataset | 82.74 | 82.74 | 82.74 | 82.74 | 82.74 | 61.27 | 61.27 | 61.27 | 61.27 | 61.27 | 72.01 |
| | ImageNet | 72.24 | 72.24 | 72.24 | 72.24 | 72.24 | 72.20 | 72.20 | 72.20 | 72.20 | 72.20 | 72.22 |

**Implementation Details of Fine-tuning Baselines.** In the main text, we compare our knowledge editing algorithm with two fine-tuning techniques. Given the limited number of samples available for each entity, we adopt parameter-efficient fine-tuning methods to maintain the generalization ability of the original model. For standard fine-tuning, we freeze all transformer layers except the last one. For LoRA (Hu et al., 2022), we set $\alpha = 32$ and $r = 8$. The optimizer used is AdamW (Loshchilov & Hutter, 2019) with a learning rate of $5 \times 10^{-6}$. Since the original CLIP model is trained with contrastive loss, but for knowledge insertion fine-tuning we only have images of a single type of object without corresponding caption data, the fine-tuning objective is defined as the cosine similarity loss between the visual representation and the text representation of the prompt "`This is a photo of [cls]`.":

$$\min_{\theta_{\text{image}}} 1 - \cos(M_{\text{image}}(I), M_{\text{text}}(T)). \tag{24}$$

Given the small sample size, overfitting to the new data remains a challenge. To ensure a fair comparison with our knowledge editing method, we save intermediate checkpoints during fine-tuning. In Table 4, we report results from the checkpoint whose category-level accuracy is closest to that of our knowledge insertion method. We observe that while fine-tuning can achieve a similar level of recognition performance, it often leads to catastrophic forgetting, significantly impairing the model's ability to generalize to other categories. In contrast, our knowledge insertion algorithm effectively preserves the generalization capabilities of the original model.

## A.8 SCALABLE KNOWLEDGE INSERTION

In the main text, we focus on single knowledge insertion. At each time, we only insert one piece of associative knowledge into the model. Here we extend the framework and show the potentials of scalable knowledge insertion. Specifically, we utilize the MEMIT method (Meng et al., 2022b), which solves the following optimization problem:

$$\min \|\hat{W}K - V\|_F^2 + \|\hat{W}K_* - V_*\|_F^2 \quad \text{by setting } \hat{W} = W + (V_* - WK_*)K_*^T(C + K_*K^T)^{-1}. \tag{25}$$

We conduct experiments on CUB-200 and Flower-101. We edit the five cases presented in the main text simultaneously. The results are shown in Table 8. This demonstrate the possibility of scaled knowledge editing, as the category-level accuracy for all cases improve significantly, with minimum degeneration of the dataset-wise accuracy and zero-shot generalization accuracy.

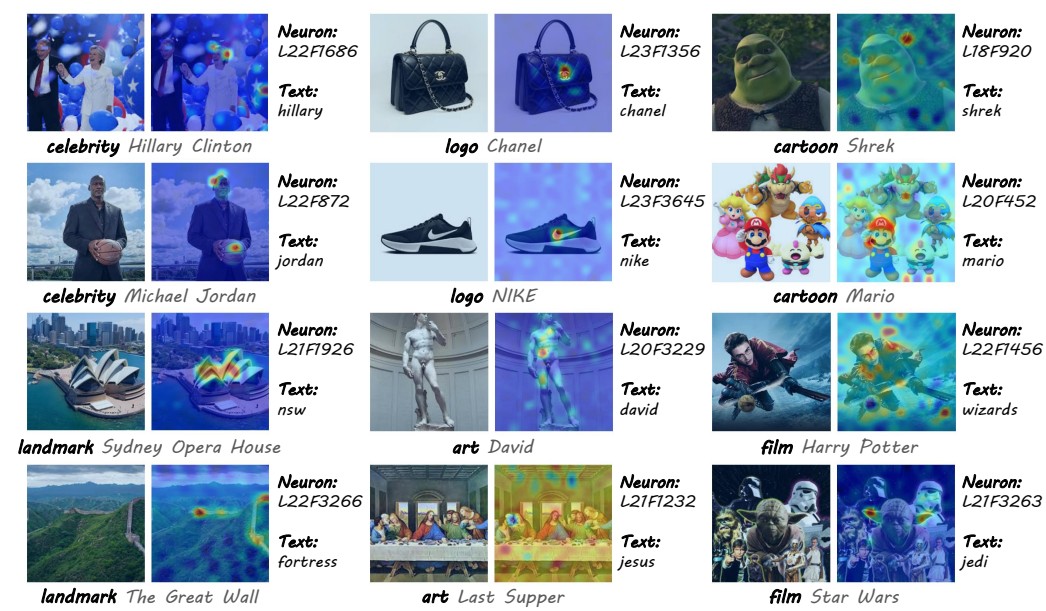

Figure 9: Illustrations of knowledge neurons and their interpretations for DFN (Fang et al., 2024).

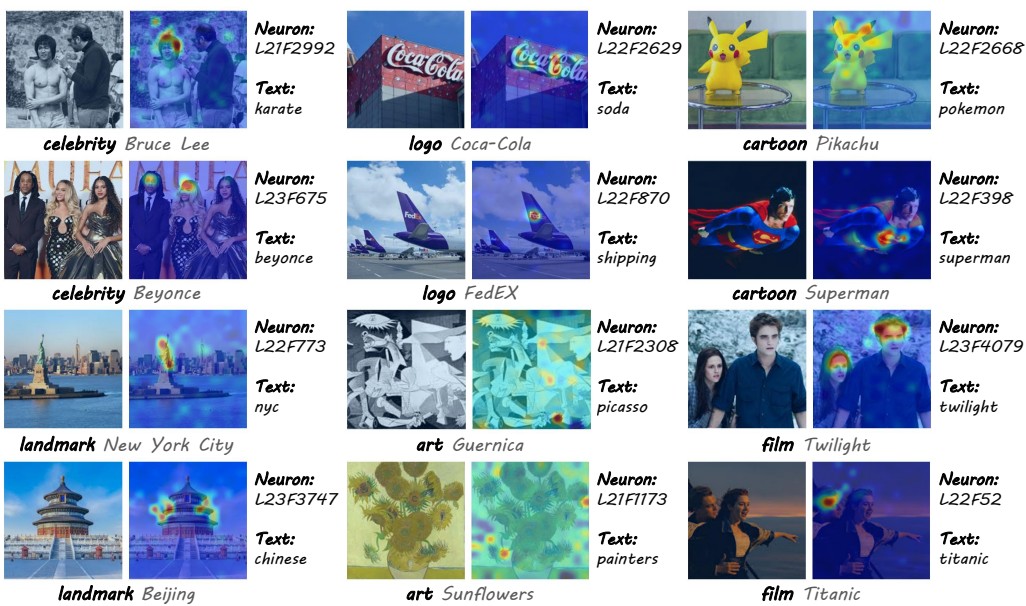

Figure 10: Illustrations of knowledge neurons and their interpretations for MetaCLIP (Xu et al., 2024).

## A.9 RESULTS OF ADDITIONAL VLMS

In this section, we examine the generalizability of the proposed neuron-attribution framework. To this end, we perform experiments on two additional vision-language models (VLMs) that utilize intermediate embedding layers for modality alignment: DFN (Fang et al., 2024) and MetaCLIP (Xu et al., 2024). Fig. 9 and Fig. 10 showcase examples of knowledge neurons identified in these models. Our results reveal that the existence of knowledge neurons is a universal phenomenon across different VLMs. Furthermore, our framework effectively identifies and interprets knowledge neurons in these models, demonstrating its strong generalizability beyond CLIP.

## A.10 LLM Usage Statement

In this work, LLMs are utilized to polish the wording and correct grammatical errors. We also use LLMs to aid coding, including debugging and generating simple functions such as data cleaning and results visualization.

