# OpenReview forum: "Towards Understanding Associative Knowledge in Vision-language Models via Neuron-level Attribution"
_ICLR.cc/2026/Conference — ICLR 2026 Conference Withdrawn Submission_

### Official Review · Reviewer_c4wP · 2025-10-27

**Soundness:** 2
**Presentation:** 3
**Contribution:** 2
**Rating:** 4
**Confidence:** 3

**Summary:**

This paper studies neuron attribution in Vision Language Models (VLMs), more specifically, CLIP. Leveraging a variant of Logit Lens, it proposes to use cosine similarity between the visual representation and the text representation as the indicator to find important “knowledge neurons”. It proposes to interpret these neurons either through text (via a sparse decomposition algorithm) or through input pixels (via image token attribution).

**Strengths:**

The paper is easy to follow.

It proposes a framework for neuron attribution, which extends previous work.

The knowledge insertion is effective, on par with or even suppressing fine-tuning-based methods for flower/bird entries.

**Weaknesses:**

- **Model limitation**. Throughout the paper, experiments are only conducted on a single model, CLIP. In principle, the framework should be applicable to all VLMs that use separate vision and text encoders (plus some kind of projectors into a shared embedding space), such as MiniGPT, Qwen-VL, etc. The lack of model diversity weakens the validity of the title, which claims studies on "vision-language models".

- **Knowledge limitation**. Though named as "knowledge neurons", the type of "knowledge" studied and covered is limited to very specific entities such as celebrities and cartoon characters, thus constraining the applicability of the proposal.

- **Many of the results are only launched on a tiny subset of neurons or manually picked subsets of images**, and thus are hard to describe as general enough to hold as scientific conclusions. For example, many subfigures in Figure 3 are only based on a few knowledge neurons; the result in Figure 4 is based on a single entry; Table 3 only contains results of 10 celebrities, not for other categories described in Section 4.1; and more.

- **Some interpretations of results are not precise**. For example, in Section 4.4 “Removal of Sensitive Knowledge”, a lower cosine similarity does not necessarily imply that the private knowledge has been completely removed. Claiming *preservation of privacy* requires more robust testing, and the evidence is not persuasive enough.

**Questions:**

- How do you obtain the numbers in Table 1, with “textual interpretation of the top attributed neurons”? Do you pick the text entry with the largest coefficient after running orthogonal matching pursuit?

- In Section 3.3, during the attribution of image patches, do you observe a similar phenomenon as [1], such that some background patches turn out to be important even though they do not carry any entity-related information?

- As for the phenomenon “Knowledge neurons suppress the activations of other neurons”, a similar (though not identical) observation has been reported as a “self-repair” mechanism in [2] [3], and it would be nice to discuss the connection to these works.

Misc: line 269, always originates from the class token, is incorrect. It contains indirect effects from the other tokens; line 456 RMOE -> ROME

References:

[1] Vision Transformers Need Registers

[2] The Hydra Effect: Emergent Self-repair in Language Model Computations

[3] Explorations of Self-Repair in Language Models

---

### Official Review · Reviewer_bg9x · 2025-10-31

**Soundness:** 2
**Presentation:** 3
**Contribution:** 2
**Rating:** 4
**Confidence:** 3

**Summary:**

This paper investigates how associative, entity-level knowledge is represented inside CLIP. It proposes a neuron-level attribution method that ranks FFN/attention units by their contribution to image–text alignment for a target entity, and shows that editing those units can remove or insert entity knowledge while largely preserving CLIP’s zero-shot performance. The authors also introduce a small evaluation set, VisEnt, tailored to entity probing.

**Strengths:**

1. The paper provides a fairly detailed analysis of “knowledge neurons” in a VLM, including the interesting finding that removing a high-contribution neuron lets other neurons’ alignment contributions increase.
2. The approach shows robustness across the tested settings (VisEnt, ImageNet slice, Flower-102/CUB insertion): edits stay mostly local and do not collapse CLIP’s zero-shot accuracy.
3. The authors create VisEnt, a purpose-built dataset for detecting specific entities, which is a useful resource.
4. On their chosen task, the method outperforms the selected baselines (activation, gradient, activation × gradient).

**Weaknesses:**

1. The study is carried out only on CLIP (ViT-L/14). Since the claim is about understanding associative knowledge in VLMs, it would strengthen the paper to repeat at least a subset of experiments on another VLM family (e.g., BLIP/BLIP-2) to show the phenomenon is not CLIP-specific.
2. The paper does not compare to widely used explanation methods such as Grad-CAM, Grad-CAM++, Score-CAM, Token-CAM, SHAP, or prototype-based methods like ProtoPNet, xDNN and other similar even though these can be evaluated on the same images with well-established protocols. If the authors would add such comparisons, I am happy to revise these scores upward and overall
score.
3. The paper mostly reports custom alignment/cosine drops, but does not report widely adopted faithfulness/confidence metrics such as Insertion/Deletion AUC and Average Drop / Increase in Confidence.
4. In Fig. 5 (App. A.5) neuron IDs are shown as L23F1520, L23A9H7, etc. The math section earlier gives the abstract forms LlFk and LlAjHk, but the figure itself does not print explicit text like “layer 23, FFN neuron 1520” next to the heatmaps. If the figures contained the actual layer + neuron description, it would be much easier to follow.

**Questions:**

Since you compare to finetuning and LoRA for knowledge insertion, can you also report the runtime / GPU memory for (i) finding the knowledge neuron(s) and (ii) applying the edit, so we can see whether multi-neuron suppression is still cheaper than LoRA?

---

### Official Review · Reviewer_KjEm · 2025-11-01

**Soundness:** 4
**Presentation:** 3
**Contribution:** 4
**Rating:** 8
**Confidence:** 4

**Summary:**

This paper investigates how associative knowledge is represented inside VLMs. It focuses on the vision encoder of CLIP. The authors propose a neuron-level attribution method to identify “knowledge neurons” that store information about specific entities such as celebrities or logos. They show that these neurons are mainly located in the final FFN layers and can be interpreted from both visual and linguistic perspectives. The paper also demonstrates practical applications, including selectively removing or inserting knowledge, providing new insights into the internal mechanisms of VLMs.

**Strengths:**

1. The paper introduces a novel neuron-level framework for analyzing associative knowledge in VLMs. It extends mechanistic interpretability to the vision encoder of CLIP. This perspective is fresh and addresses an under-explored aspect of multimodal learning.
2. The methodology is rigorous and carefully designed. The authors systematically quantify neuron-level contributions, validate findings across multiple model variants, and support their analysis with both qualitative visualizations and quantitative ablation studies. It lends strong credibility to the results.
3. The experimental design is thorough and convincing. It covers layer-wise analysis, ablation studies, and visualization of neuron behavior.

**Weaknesses:**

1. Experiments focus mainly on recognizing specific entities (e.g., celebrities, logos). It is still unclear whether the method generalizes to other knowledge types like relational or compositional concepts. Broader task evaluation would strengthen the claims.
2. The paper only studies CLIP-style dual-tower models and does not explore other types of multimodal architectures. It limits the generality of the conclusions.

**Questions:**

Could the proposed framework identify neurons responsible for non-entity knowledge, such as relational or compositional concepts？

---

### Official Review · Reviewer_UWbb · 2025-11-08

**Soundness:** 1
**Presentation:** 1
**Contribution:** 1
**Rating:** 2
**Confidence:** 4

**Summary:**

This paper investigates how CLIP stores associative knowledge about specific entities. The authors develop neuron-level attribution methods to identify "knowledge neurons" within CLIP's visual encoder and empirically verify that the recognition of entities is primarily facilitated by a small subset of neurons in the FFN modules in the later layers. They also propose knowledge editing methods including selectively removing sensitive knowledge and inserting new entity associations without degrading overall performance.

**Strengths:**

The paper provides a comprehensive empirical analysis of neuron distributions across layers, demonstrating that entity-specific knowledge neurons concentrate in later FFN layers. Their intervention experiments also reveal interesting dynamics in how knowledge neurons interact.

**Weaknesses:**

1. The Introduction fails to clearly illustrate the motivation and the paper's main claims. Specifically, the second paragraph reads as a superficial survey that lacks logical coherence. While it provides interesting well-known background information about CLIP, it doesn’t specify why understanding this mechanism is significant and how limited the current landscape is. I would expect the introduction to elaborate on why understanding this mechanism matters and what practical implications it has, rather than appearing to pursue interpretation for its own sake, like “interpreting for interpretability”. Moreover, I also find the last sentence in the third paragraph vague and sort of over-generalizing. Knowledge associations in CLIP and VLMs are technically different problems. The framing of 'isolating the visual encoder to avoid ambiguity' is misleading—the paper simply studies CLIP specifically rather than general VLMs.

2. Overstated novelty claims. The paper claims to 'pioneer the investigation of associative knowledge neurons in VLMs', which is factually incorrect. Goh et al. (2021) [1], cited in the paper itself, explicitly studied neurons in CLIP's vision encoder that respond to specific entities including celebrities and fictional characters. While this work may contribute new methodological approaches, the fundamental investigation of entity-specific neurons in CLIP is not novel.

3. Missing citations. To my knowledge, the equation (3) is equivalent to the attention transformation in Anthropic’s report back to 2021 [2]. Also, around equation (4) and (5), this style of logit attribution methods is already proposed by previous works, like [3].

4. The experiments warrant little external validity. The empirical evidence of the superiority of the proposed attribution methodology relies on the curated dataset the authors crafted for this work.

5. Section 5 is overly perfunctory. This paper seems to be not ready for submission, especially when I read the final section, and it could potentially hurt the reputation of the authors and their teams.

6. No source code or instructions for reproducibility is provided.

Overall, the paper needs careful revision on its presentation and the credibility is low in the form as is.

[1] Goh, Gabriel, et al. "Multimodal neurons in artificial neural networks." Distill 6.3 (2021): e30.

[2] Elhage N, Nanda N, Olsson C, et al. A mathematical framework for transformer circuits[J]. Transformer Circuits Thread, 2021, 1(1): 12.

[3] Wang, Kevin, et al. "Interpretability in the wild: a circuit for indirect object identification in gpt-2 small." arXiv preprint arXiv:2211.00593 (2022).

**Questions:**

See Weaknesses. Also,

1. The term Associative Knowledge can be sensible in the context of CLIP, but is it an established jargon from human learning theories or other research? If it is a term for the illustration of this paper, I’d suggest carefully defining it somewhere in the Intro.

2. Which evidence could support the claim made in Introduction that “knowledge neurons are integral to recognizing these specific entities”? The knowledge removal experiment indicates that CLIP can still maintain decent knowledge after masking part of the identified neurons.

---

### Note · Authors · 2025-11-14

I have read and agree with the venue's withdrawal policy on behalf of myself and my co-authors.